A needle in a haystack: Mesozoic origin of parasitism in Strepsiptera revealed by first definite Cretaceous primary larva (Insecta)

Pohl Hans hans.pohl@uni-jena.de 1
Batelka Jan 2
Prokop Jakub 2
Müller Patrick 3
Yavorskaya Margarita I. 1
Beutel Rolf G. 1
1 Institut für Zoologie und Evolutionsforschung, Spezielle Zoologie und Entomologie, Friedrich-Schiller Universität Jena , Jena , Germany
2 Department of Zoology, Faculty of Science, Charles University Prague , Praha , Czech Republic
3 Käshofen , Germany
De Baets Kenneth
Electronic publication date: 2018 Nov 22
Publication date: 2018
Volume: 6
Electronic Location ID: e5943
Received 2018 Aug 15; Accepted 2018 Oct 10
Copyright: ©2018 Pohl et al.
Copyright year: 2018
Copyright holder: Pohl et al.
License: This is an open access article distributed under the terms of the Creative Commons Attribution License, which permits unrestricted use, distribution, reproduction and adaptation in any medium and for any purpose provided that it is properly attributed. For attribution, the original author(s), title, publication source (PeerJ) and either DOI or URL of the article must be cited.
License URL: https://creativecommons.org/licenses/by/4.0/

Keywords: Burmite, Evolutionary stasis, Larva, Cretaceous, Strepsiptera

Funding: European Regional Development Fund State budget of the Czech Republic CZ.1.05/4.1.00/16.0347 Institutional Research Support grant of the Charles University, Prague SVV 260 434/2018 Charles University Grant Agency 1546218 The publication is co-financed by the European Regional Development Fund and the state budget of the Czech Republic (project No. CZ.1.05/4.1.00/16.0347). The work of Jan Batelka was supported by the Institutional Research Support grant of the Charles University, Prague (No. SVV 260 434/2018) and by the Charles University Grant Agency (GAUK, No. 1546218). The funders had no role in study design, data collection and analysis, decision to publish, or preparation of the manuscript.

==============================
Twisted winged insects (Strepsiptera) are a highly specialized small order of parasitic insects. Whether parasitism developed at an early or late stage in the evolution of the group was unknown. Here we record and describe the first definite Mesozoic strepsipteran primary larva embedded in Burmese amber (∼99 million years ago). This extends the origin of parasitism back by at least ∼50 million years, and reveals that this specialized life style has evolved in the Mesozoic or even earlier in the group. The extremely small first instar displays all diagnostic characters of strepsipteran immatures of this stage and is nearly identical with those of Mengenillidae, one of the most “ancestral” extant strepsipteran taxa. This demonstrates a remarkable evolutionary stasis over  100 million years. The new finding strongly weakens the case of small larvae embedded in Cretaceous amber interpreted as strepsipteran immatures. They differ in many structural features from extant strepsipteran primary larvae and are very likely parasitic beetle larvae.

Introduction

Strepsiptera is a highly specialized small order of holometabolous insects (Pohl & Beutel, 2008; Pohl & Beutel, 2013). The phylogenetic placement of the group, one of the longest controversies in systematic entomology, was only recently clarified (Niehuis et al., 2012; Peters et al., 2014; Misof et al., 2014; see also Kjer et al., 2016). The oldest recorded fossils are known from Cretaceous Burmese amber (Grimaldi, Kathirithamby & Schawaroch, 2005; Pohl & Beutel, 2016; Engel et al., 2016), indicating a minimum age of ca. 100 mya. An age of origin of ca. 120 mya was estimated based on molecular data (McMahon, Hayward & Kathirithamby, 2011). However, the confirmed sister group relationship with Coleoptera (Misof et al., 2014) implies that the group originated already in the early Permian or late Carboniferous (McKenna et al., 2015; Toussaint et al., 2017).

Extant strepsipterans are characterized by a conspicuous sexual dimorphism. The winged males are free living but extremely short-lived. With well-developed sensory organs and a highly efficient flight apparatus they are able to find the females within their very short life span of only few hours. Females are wingless and morphologically strongly simplified. They develop as endoparasites of other insects. Only the females of the basal extant Mengenillidae (and probably Bahiaxenidae) leave their host and are able to move actively. The females of the majority of the species (Stylopidia) are permanently endoparasitic and only penetrate the host’s abdominal intersegmental membranes with the sclerotized anterior part of their body (Pohl & Beutel, 2008). The primary larvae of Strepsiptera are among the smallest known metazoans with an average size of ca. 230 µm, comparable to unicellular ciliates of the genus Paramecium (Pohl, 2002). This size reduction enables the female to produce a huge number of offspring and the minute primary larvae are able to penetrate relatively small insect hosts such as Delphacidae (Auchenorrhyncha) with an adult size of 1.5–6 mm (Pohl & Beutel, 2008). The body of the primary larvae is elongated oval to nearly drop shaped. The head is semicircular with well-developed stemmata with cornea lenses. Antennae are missing and the labrum is not present as a separate element. Sternal plates inserted between the coxae with spinulae and fringes of microtrichia on the posterior border of the abdominal sternites produce capillary forces enabling the larvae to stick to and crawl along wet surfaces (Pohl & Beutel, 2008). A unique apomorphic feature within Holometabola is the presence of one pair of long and strongly developed bristle-like cerci. They are inserted on the terminal abdominal segment XI, with strong muscles attached to their base. They enable the minute larvae to jump very efficiently, an ability secondarily lost in the primary larvae of the Stylopidae (Pohl & Beutel, 2005).

The relatively sparse Cretaceous fossil record of adult strepsipteran males suggests that they were already a specialized group of insects in the late Mesozoic. It is very likely that the highly modified antennae with flabellate appendages and numerous specialised dome-shaped chemoreceptors were used for finding females over a relatively large distance as in extant species of the group. Likewise, the uptake of food was apparently very limited at least, as suggested by the strongly simplified mouthparts. Strepsipteran females from the Mesozoic have not been discovered yet. As with most cases of parasitism documented in the fossil record (e.g., Nagler & Haug, 2015), Mesozoic Strepsiptera provide only indirect evidence of this specialization. Based on the small size of adult males (ca. 1.5–3 mm), it was assumed, that the species were already endoparasites of other insects (Pohl & Beutel, 2008). However, conclusive evidence by direct detection of definite strepsipteran larvae was lacking, and therefore direct evidence for an endoparasitic life style of the immature stages of Mesozoic strepsipterans. Previously, endoparasitism was only documented by fossil representatives of Stylopidia from Eocene deposits, the strepsipteran subgroup with permanently endoparasitic females (Kinzelbach & Pohl, 1994; Henderickx et al., 2013).

Confirmed fossil records of strepsipteran primary larvae are from the Eocene and Miocene, respectively. The oldest fossil from Eocene brown coal (Geisel Valley, Germany) was initially described as a first instar of a scale insect (Coccoidea) (Haupt, 1950). It was later assigned to the extant strepsipteran genus Stichotrema (Myrmecolacidae) (Kinzelbach & Lutz, 1985). Finally, it was placed as Stylopidia incertae sedis based on a re-examination and a cladistic analysis of characters of primary larvae of all extant families of Strepsiptera (Pohl, 2009). Strepsipteran primary larvae associated with its parent female and its host (Auchenorrhyncha: Delphacidae) are reported from Dominican amber (Poinar, 2004), which is currently attributed to the Miocene: 15–20 mya (Iturralde-Vinent & MacPhee, 1996; Iturralde-Vinent, 2001). Other very small “triungulin” larvae assigned to Strepsiptera were described from the Late Cretaceous (Campanian) amber of Manitoba, Canada (Grimaldi, Kathirithamby & Schawaroch, 2005), and a “planidium” from the Upper Cretaceous (Santonian) amber of the Taimyr Peninsula, Siberia (Kathirithamby et al., 2017). The interpretation of the immatures treated in the earlier study was discussed critically, pointing out an entire series of features in conflict with an assignment to Strepsiptera (Beutel et al., 2016). The “planidium” from the Upper Cretaceous amber (Kathirithamby et al., 2017) is most likely a parasitic beetle larva based on the following features: small size (ca. 0.53 mm), conical head, absence of distinctly developed stemmata, coarse ctenidia, large pretarsal adhesive pad, absence of abdominal segment XI and terminal bristles (Batelka et al., 2018) (see below).

Kathirithamby et al. (2017) used the term “planidium” for primary larvae of Strepsiptera. However, this is only appropriate for legless larvae of parasitic Diptera or Hymenoptera (Askew, 1971; Stehr, 1991). Triungulinid was introduced by Pierce (1909) for first instars of Strepsiptera based on their similarity with primary larvae of Meloidae or Ripiphoridae (Coleoptera), which were addressed as triungulins. However, as the structural affinities are only superficial and obviously non-homologous, and the first instars differ in important features (e.g., nine abdominal segments in Meloidae and Ripiphoridae versus 11 in primary larvae of Strepsiptera), we prefer the neutral term primary larva for first instars of Strepsiptera. Claws forming a trident with spatulate setae occur in phoretic primary larvae of some genera of Meloidae (Bologna, Turco & Pinto, 2010), but are completely lacking in strepsipteran larvae.

In the present study we describe a minute larva from a piece of Burmese amber, with a habitus and a set of observable features unambiguously confirming a placement in Strepsiptera. The ordinal assignment among Strepsiptera is discussed. Based on the described features the position of other putative strepsipteran larvae (Grimaldi, Kathirithamby & Schawaroch, 2005; Kathirithamby et al., 2017) is critically re-evaluated. The new fossil larva clearly confirms that Mesozoic Strepsiptera were already endoparasites of other insects. The origin of endoparasitism is extended back by ca. 50 million years to a minimum of ca. 100 Ma.

Material & Methods

Material

The piece of Burmese amber with the strepsipteran larva came from deposits in the Hukawng Valley of Myanmar. The age is estimated as ca. 99 Ma (earliest Cenomanian; Shi et al., 2012).

The amber with the strepsipteran first instar (accession number BU-002386) is integrated into the collection of the Institute of Zoology, Chinese Academy of Sciences (Beijing, P.R. China) and will be deposited in the Three Gorges Entomological Museum, Chongqing, China after 2027. It is from a mining locality at Noije Bum (near Tanai Village, 26°21′33.41″N, 96°43′11.88″E) (Cruickshank & Ko, 2003; Grimaldi, Engel & Nascimbene, 2009). The size of the piece of amber is 26 × 22 × 10 mm. Syninclusions are listed in the following: Acari (nine specimens), orthopteran nymph (1), Psocodea (1), Sternorrhyncha (1), apocritan Hymenoptera (2), Berothidae (Neuroptera) (1), Elateridae (1), primary longipedes larvae of Ripiphoridae (46), Polyphaga with unclear affinity (2), “nematoceran” species of Diptera with unclear affinity (2), brachyceran species of Diptera with unclear affinity (2).

As important syninclusions are embedded very close to the strepsipteran larva, it was not possible to isolate the larva and trim the amber piece into a thin plate and mount it on a glass microscope slide. This precludes examining the first instar with oil-immersion lenses and phase or differential interference contrast at a magnification of 1,000×. Therefore, some structures of the minute larva could not be evaluated.

Specimen imaging

The piece of amber was temporarily mounted on coverslips using glycerine. The specimen was observed under two different microscopes: An Axio Zoom.V16 with a PlanNeoFluar Z 1.0x (Carl Zeiss Microscopy GmbH) was used for the overview images and the images were saved as CZI files. For observations and for measurements ZEN 2.3 lite (blue edition) (Carl Zeiss Microscopy GmbH) was used. For higher magnifications, an Olympus IX81 inverted fluorescence microscope with UIS2 objectives, equipped with an ORCA-AG monochromatic 12-bit CCD camera (Hammatsu) was used. The mirror images were superimposed with Cell ˆR software (Olympus Soft Imaging Solutions). Sets of photographs were analyzed with Fiji (Schindelin et al., 2012).

Single images were exported with ZEN 2.3 lite or Fiji respectively. Some images were combined with Zerene Stacker (Zerene Systems LLC, Richland, USA). The photographs were processed using Adobe Photoshop® CS6 (Adobe System Incorporated, San Jose, CA, USA) and arranged as plates. Adobe Illustrator® CS6 (Adobe Systems Incorporated, San Jose, CA, USA) was used for the lettering of the plates. Image stacks of the Olympus IX81 microscope were used for the drawings and description.

Comparative taxonomy and terminology

Data on the morphology of Strepsiptera larvae are taken from (Pohl, 2000; Pohl, 2002) and the morphological nomenclature used in these studies is applied. Additionally, primary larvae of Eoxenos laboulbenei Peyerimhoff, 1919 and Mengenilla chobauti Hofeneder, 1910 (both Mengenillidae) were examined (research collection of H.P. at Phyletisches Museum). For comparison primary larvae of E. laboulbenei embedded in Canada balsam on glass microscope slides were examined with the same microscope (Olympus IX81) as the fossil.

Results

Preservation

Strongly depressed dorsoventrally, dorsal and ventral side thus difficult to distinguish, especially in abdominal region. Part of left side of head and thorax covered by debris (Fig. 1A).

Figure 1 Strepsiptera primary larva in Burmese amber, ventral view.

(A) Photomicrograph with an Axio Zoom.V16 with a PlanNeoFluar Z 1.0x. (B) Drawing based on photomicrographs with an Olympus IX81 inverted fluorescence microscope with UIS2 objective. Abbreviations: af, antennal field; cb, caudal seta; cx, coxa; fe, femur; fs, frontal seta; lcb, lateral caudal seta; mp, maxillary palp; mssp, mesosternal plate; mt, metanotum; mx, maxilla; prsp, prosternal plate; sbsIX/X, segmental border between abdominal sternites IX/X; sbtVIII/IX, segmental border between abdominal tergites VIII/IX; sI–sIX, abdominal sternites I–XI; st, stemmata; ta, tarsus; te, tentorium; ti, tibia; X, abdominal segment X; XI, abdominal segment XI.

Morphology

First instar extremely small, total length excluding terminal bristles 197 µm. Head semicircular, with recognizable stemmata but lacking antennae. Sides of body subparallel, slightly convex. Dorsum smooth, without recognizable surface structures and largely devoid of setae. Terminal abdominal segment XI with two pairs of strongly developed bristles. Specimen preserved in ventral position.

Exposed part of head capsule semicircular in ventral view, with evenly rounded anterior margin and greatest width at hind margin. Visible part distinctly shorter than maximum width. Six large individual stemmata with cornea lenses recognizable posterolaterally on left side in strongly pigmented area, only three visible on right side (st in Figs. 1A, 1B and 2A). Labrum not present as separate element. Evenly rounded anterior margin of head capsule apparently forming sharp edge, lacking median emargination (Figs. 1A, 1B, 2A and 2B). Antennae not recognizable as prominent structures, largely reduced. Antennal field likely represented by small circular structure adjacent to anteriormost stemma, visible through translucent cuticle (af in Figs. 1B and 2A). Mandibles not visible. Ventral side of head medially covered by strongly modified maxillae and labium. Maxillae medially fused, forming slightly curved, transverse plate-like structure (Figs. 1B and 2B). Maxillary palps recognizable as circular spots on posterior maxillary margin (mp in Figs. 1B and 2B). Details of labium not visible. Paired anterior tentorial arms visible in posterolateral cephalic region (te in Figs. 1B and 2B). Ecdysial sutures not visible. One seta recognizable on dorsal surface (fs in Fig. 1B). Cuticle without recognizable surface modifications.

Figure 2 Strepsiptera primary larva in Burmese amber in comparison with an extant primary larva.

(A, B) Strepsiptera primary larva in Burmese amber, head, pro-, and mesothorax, ventral view. (C, D) Eoxenos laboulbenei, head, pro-, and mesothorax, ventral view. Photomicrographs with an Olympus IX81 inverted fluorescence microscope with UIS2 objective. Abbreviations: af, antennal field; cos, coxal seta; lb, labium; mp, maxillary palp; mssp, mesosternal plate; mx, maxilla; prsp, prosternal plate; st, stemmata; te, tentorium; ti, tibia.

Pro-, meso- and metathorax subequal in length on ventral side. Only inflicted lateral margin of nota visible (Fig. 1). Thoracic segments continuously widening from anterior to posterior. Prothorax with one seta on lateral pronotal margin, visible on right side. Conspicuous sternal plate inserted between coxae (prsp in Figs. 1B and 2A). Anterior part of sternal plate broad, posterior part with triangular apex. Spinulae on posterior margin not visible. Meso- and metathorax very similar to prothorax, but sternal plate only visible on the former. Legs only partly visible, short, composed of large, transverse coxa, trochanterofemur, tibia, and one-segmented pad-like tarsus (Fig. 1B). Tarsus without claws. Hind leg with long seta on mesal margin of coxa.

Abdomen composed of eleven segments. Segment II broadest, following segments slightly tapering posteriorly (Fig. 1). Only lateral margins of tergites visible, with two lateral setae on most of them (Figs. 1B, 3A and 3B). Sternites I–IX half as long as thoracic segments. One short seta visible on right side of sternite III, inserted close to hind margin. Sternites IV–VI with two setae inserted laterally close to hind margin (Fig. 1B). Very fine parallel longitudinal lines on margins of sternites I–VIII may represent spinulae, or alternatively tergal furrows visible through cuticle (Fig. 3A). Tergites IX and X fused, about as long as segments VI–VIII combined, forming large plate-like structure covering terminal segment XI (Fig. 4A). Hind margin of segment XI truncated, with pair of very strongly developed bristles inserted close to midline, and second similar but shorter pair more laterally (Figs. 1B, 3B and 4B).

Figure 3 Strepsiptera primary larva in Burmese amber in comparison with an extant primary larva.

(A, B) Strepsiptera primary larva in Burmese amber. (A) Meso-, metathorax, and anterior abdominal segments, ventral view. (B) Terminal abdominal segments. (C, D) Eoxenos laboulbenei. (C) Abdominal segments III–IX, ventral view. (D) Terminal abdominal segments. Photomicrographs with an Olympus IX81 inverted fluorescence microscope with UIS2 objective. Abbreviations: cb, caudal bristle; lcb, lateral caudal bristle; sbsIX/X, segmental border between sternite IX and X; sbsX/XI, segmental border between sternite X and XI; sbtVIII/IX, segmental border between tergite VIII and IX; ti, tibia; X, abdominal segment X; XI, abdominal segment XI.

Figure 4 Terminal segments of Strepsiptera primary larva in Burmese amber, drawings based on photomicrographs with an Olympus IX81 inverted fluorescence microscope with UIS2 objective.

(A) Dorsal view. (B) Ventral view. Abbreviations: sbsIX/X, segmental border between abdominal sternites IX/X; sbtVIII/IX, segmental border between abdominal tergites VIII/IX; X, abdominal segment X; XI, abdominal segment XI.

Diagnosis

Differs from primary larvae of Eoxenos laboulbenei by the presence of six stemmata and from primary larvae of Mengenilla chobauti (both Mengenillidae) and all other known primary larvae of Strepsiptera by the strongly developed second pair of bristles on the hind margin of segment XI.

Discussion

The larva can be easily and unambiguously assigned to Strepsiptera, based on several diagnostic features and also apomorphic character states shared with other first instars of the order. Like extant strepsipteran primary larvae (Pohl, 2000) the fossil has an elongated oval to drop shaped body, with a semicircular head with large stemmata with cornea lenses, but lacking a separate labrum and also antennae as visible prominent structures. One pair of long and strongly developed bristles inserted on the terminal abdominal segment XI is present (Figs. 1, 5A and 5B). Within Strepsiptera, the larva shows close structural affinities with first instars of Eoxenos laboulbenei, especially due to the identical equipment with bristles on the last abdominal segments (Figs. 4 and 5). A placement close to the root of Strepsiptera s.l. or s.str. is likely. However, as only the male adults of extant Bahiaxenos and the extinct †Mengea, †Protoxenos (both Eocene Baltic amber), †Cretostylops, †Kinzelbachilla, and †Phthanoxenos, (Cretaceous Burmese amber) are known (Grimaldi, Kathirithamby & Schawaroch, 2005; Pohl, Beutel & Kinzelbach, 2005; Bravo et al., 2009; Pohl & Beutel, 2016; Engel et al., 2016), a precise phylogenetic assessment is not possible. It is conceivable that the larva belongs to one of the three strepsipteran species known from the same fossil site, but a verification is not possible with the information at hand.

Figure 5 Primary larva of Eoxenos laboulbenei, drawings based on scanning electron micrographs (modified from Pohl (2000)).

(A) Ventral view. (B) Dorsal view. Abbreviations: af, antennal field; cb, caudal bristle; cx, coxa; fe, femur; fs, frontal seta; lb, labium; lcb, lateral caudal bristle; mp, maxillary palp; mx, maxilla; prsp, prosternal plate; sI–sXI, abdominal sternite I–XI; st, stemma; ta, tarsus; ti, tibia; tI–tX, abdominal tergite I–X.

A major point demonstrated here is that this Cretaceous primary larva of Strepsiptera differ only in minimal details from extant immatures of basal genera of the order (Mengenillidae). This documents a high evolutionary stability over ca. 100 million years and clearly suggests that these extremely miniaturized larvae were already parasitic and produced in high numbers.

Large stemmata for identification of a host and the abdominal jumping apparatus are features clearly linked with parasitism, and the extremely small size suggests a very high number of offspring like in extant groups (Pohl & Beutel, 2008).

The finding of a Cretaceous primary larva nearly identical with those of the extant genus Eoxenos sheds new light on recently described “planidia” assigned to Strepsiptera (Kathirithamby et al., 2017). Considering the clearly demonstrated evolutionary stability of the tiny first instars, it appears highly unlikely that at the same time aberrant and unusually large strepsipteran primary larvae occurred. The “planidia” described by Kathirithamby et al. (2017) differ in many features from strepsipteran larvae, such as for instance much larger size (ca. 0.5 mm), anteriorly conical head, absence of large stemmata (“presumably two stemmata”), presence of a membranous cervix, presence of ctenidia, posteriorly widening meso- and metanota, absence of sternal plates, strongly widened femora, lack of setae and spines on the abdominal sternites, only ten abdominal segments, lacking terminal bristles (on segment XI in Strepsiptera), and consequently the lack of a jumping apparatus. It is much more likely that these were simply miniaturized parasitic beetle larvae (Beutel et al., 2016), although their family placement is still open to debate (Batelka et al., 2018). All observed features are compatible with a placement in polyphagan beetles. The presence of coarse ctenidia and a pad-like pretarsal adhesive device are apomorphic features linking it clearly with beetle larvae very likely belonging to the cucujiform family Ripiphoridae (Beutel et al., 2016; Batelka et al., 2018).

Considering the large number of primary larvae produced by strepsipteran females of Mengenillidae (ca. 2,500 by a 5 mm large female of E. laboulbenei (Silvestri, 1941)), it is surprising at first glance that not more strepsipteran primary larvae have been found in amber. The larva we describe here was only found by chance, like the primary larva from Eocene brown coal (Voigt, 1938). Due to their extremely small size, the larvae are only visible with high magnification (100×) and can be overlooked very easily. In contrast to primary longipedes larvae of Ripiphoridae (Batelka et al., 2018), extant primary larvae of Strepsiptera show no aggregation behaviour (H Pohl, pers. obs., 2018). After emerging from the female, the larvae quickly disperse in search of a suitable host. Considering their very small size, their radius of activity is probably limited to a few meters.

Conclusions

Detailed investigations of Burmese amber revealed the first definitive strepsipteran primary larva from the Cretaceous. Diagnostic features are the size of less than 200 µm, an elongated oval body, a semicircular head, and stemmata with cornea lenses in a strongly pigmented area. Apomorphic character states shared with other primary larvae of Strepsiptera are the lack of a separate labrum and prominent antennae, the medially fused maxillae, sternal plates, and the presence of a pair of long and strongly developed bristles inserted on the terminal abdominal segment XI. An evolutionary stability over ca. 100 million years is revealed as the Cretaceous primary larva of Strepsiptera differs only in minimal details from extant Mengenillidae, the sistergroup of the vast majority of the Strepsiptera (ca. 97% of the species). The recently described “planidium” assigned to Strepsiptera by Kathirithamby et al. (2017) differs in many characters from fossil and extant primary larvae of Strepsiptera and is very likely a parasitic beetle larva.

We are grateful to Mr. Weiwei Zhang for providing pieces of amber used in the context of another publication, and to Assoc. Prof. Ming Bai for arranging this loan. The authors thank Ondřej Šebesta (Charles University) for his help with the examination setup on the Olympus IX81 and Birgit Perner (Leibniz-Institut für Alternsforschung—Fritz-Lippmann-Institut, Jena) for providing us access to the Axio Zoom.V16. We also thank Benjamin Naumann (Friedrich-Schiller-Universität Jena) for his help with the Axio Zoom.V16. Finally, we would like to thank Kenneth De Baets, Kateřina Votýpková and one anonymous reviewer for their helpful suggestions, which improved the manuscript.

Additional Information and Declarations

Competing Interests

Author Contributions

Data Availability

The authors declare there are no competing interests.

Hans Pohl conceived and designed the experiments, performed the experiments, analyzed the data, contributed reagents/materials/analysis tools, prepared figures and/or tables, authored or reviewed drafts of the paper, approved the final draft.

Jan Batelka, Margarita I. Yavorskaya and Rolf G. Beutel analyzed the data, authored or reviewed drafts of the paper, approved the final draft.

Jakub Prokop performed the experiments, analyzed the data, contributed reagents/materials/analysis tools, authored or reviewed drafts of the paper, approved the final draft.

Patrick Müller approved the final draft.

The following information was supplied regarding data availability:

Pohl, Hans; Batelka, Jan; Prokop, Jakub; Müller, Patrick; Yavorskaya, Margarita; Beutel, Rolf (2018): First definite Cretaceous Strepsiptera primary larva (Insecta). figshare. Collection. https://doi.org/10.6084/m9.figshare.c.4196174.v1.

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
