# Peer review of "A needle in a haystack: Mesozoic origin of parasitism in Strepsiptera revealed by first definite Cretaceous primary larva (Insecta)"

_PeerJ, doi:10.7717/peerj.5943_

## Round 0.1 · original submission · Minor Revisions

This is a nice description of the first unambiguous Cretaceous strepsipteran immature. I would love to see this published, but there are still some minor, but crucial points I would like to see addressed before publication.

The main points are:

Direct versus indirect evidence of parasitism: you present an important case of direct evidence for parasitic immatures. The general importance of direct evidence rather than indirect evidence for inferring parasitic life stages has been discussed before (e.g., Nagler & Haug 2015). I feel it would benefit the reader by highlighting that presence of parasitic larval stages is often inferred based on the present of adult representatives (not just in Strepsiptera; see Nagler and Haug, 2015) and not by direct finds of parasitic larval stages making this an exceptional find.

Bearing on interpretation of previously reported putative strepsipteran immatures: I agree that these are the first unambiguous crown-group strepsipteran immatures. However, I miss a discussion on what a stem-group strepsipteran immature could have looked like. I concur that this is hard to do, but at least some informed suggestions can be made. You can potentially speak of stasis in this lineage – although ideally one would have multiple samples / taxa demonstrating this to be since their inception. How can you rule out that different strategies or intermediate morphologies (with that seen in their sister-lineage) not existed in early representative or now-extinct lineages of Strepsiptera ? Particularly, since this group has an ancient origin inferred to be traced back to the Late Carboniferous or Early Permian... Morphological intermediate lineages could be extinct today. Based on the discovery of a single immature, you can only make limit inference on other larval stages present in this interval. The rejection of previous interpretation and re-assignment should be based on characters and is most likely discussed in other articles. I am sure these are trivial to you, but it is currently not obvious to non-expert readers and should be at least briefly discussed. Does it share more characters with parasitic beetles than with Strepsiptera? How much do we know about larvae across both sister lineages. Even so can you rule out that this state represented in beetles could be ancestral for Strepsiptera? I therefore feel you should rephrase these aspects and express yourself more carefully relating to some of these aspects. Some suggestions are made in the annotated pdf.

Please address the comments of the reviewers and those listed in the annotated pdf, in addition to these points.

Suggested reference:
Nagler C, and Haug JT. 2015. From Fossil Parasitoids to Vectors: Insects as Parasites and Hosts. Advances in Parasitology 90:137-200.

Reviewer 1 ·

Basic reporting

The study is thorough and generally the conclusions are appropriate. There are numerous minor edits needed in sentence structure and grammar. For example on line 62 and 63 the last word “yet” should be removed. This is just an example of many minor changes needed to improve the writing of the paper.

Experimental design

no comment

Validity of the findings

• There is little doubt that this specimen constitutes a strepsipteran primary larvae and therefore the interpretation of this specimen is important to science.
• It is valuable that the authors discuss and correct the use of larval terms for the first instar larvae of Strepsiptera. The terms planidium and triungulin are obviously not appropriate for Strepsiptera larva but have been repeatedly used incorrectly.
• In lines 155-156, there is no need to note that wings or wing buds are missing. No evidence of these would be expected in a strepsipteran larvae or any larvae that might be mistaken for this order, such as Meloidae, Rhipiphoridae, …
• The term hatching in line 244 in not appropriate. The larvae emerge from the female after hatching internally from an egg. The term in this sentence should be changed to emerging.
• In more than one place in the manuscript it is suggested that there must be a large number of primary larvae produced by each female but this appears to be speculation and if included should be identified as such. Based on knowledge of current strepsipterans, this could be true but the authors should not rely so heavily on this interpretation as fact. The reality is that we know nothing of the natural history of these larvae or their numbers from this geological age. Unless the unlikely event of the discovery of a gravid female of this age it may always be speculation as to the relative number of offspring. Likewise, lines 243-246 are stated as fact but is speculation for the species covered in this study.

Additional comments

This is a valuable paper that appears to only need minor changes. My suggestions are offered to try to improve an already generally good paper.

·

Basic reporting

The manuscript fully meets the standards and is within the scope of the journal. It provides sufficient backround and introduction to place the work into the broader field of knowledge.The text is is written in professional, unambiguous language, so an international audience can clearly understand. The sources are adequately and appropriately cited. Very important part of manuscript are the figures, which are well described, in a high quality and sufficient resolution (as much as possible). The text is organized into coherent subsections.
The results are definitely ‘self-contained’, while results are relevant to the hypothesis.

Experimental design

It is clearly stated how research fills and identified knowledge gap. The investigation was conducted to very high technical standard, the methods are comprehensive and described sufficiently to be reproducible.

Validity of the findings

Conclusion are limited to supporting results and are appropriately stated. Speculations are identified as such.

Additional comments

According to my opinion, there are a few notes that need more details or to be slightly modified, usually to ensure that an audience can clearly understand the text. I cannot find any serious weakness.

Line 19: Mesozoic origin – it could be „at least“ the Mesozoic origin
Line 21: „one of the most“ or the most?
Line 42: Bahiaxenidae probably also leave their host – is it supported by references?
Line 50: It is not clear, whether labrum is completly missing, or it is „largely reduced“ (written in Lines 163-166) / or fused with other mouthparts – „lacking separate labrum“ (Lines 204, 205), so I would prefer "not present as separate element".
Line 56: „enable the minute larvae to jump very efficiently“ – not true for the family Stylopidae, so I would kindly suggest "for the most" of families"
Line 84: I am not familiar with an abbreviation „s.b.“
Line 85-87: Is the „planidium“ always „legless“? Aren’t also first instar larvae of Meloidae (with legs) reffered as „planidium“? This term is derived from the Greek word, which should mean „wanderer“, so I assume the term is not such a problem in case of Strepsiptera.
Lines 87-94: Although I agree with neutral term „primary larva“ for first instars of Strepsiptera, term „triungulinid“ (= triungulin-like) also would not be such a problem for case of Strepsiptera, because it is not the same as „triungulin“ (without -id).
Line 101: I would suggest to add the final years again – "endoparasitism is extended back by ca. 50 milion years to minimum age of ca. 100 mya"
Line 106: Better to use mya consistantly than „Ma“
Line 163: (st in Figs. 1A, B, 3A)
Line 187: Sternites IV-VI with two setae, not Sternites III-IV
Line 197: Mengenilla chobauti
Line 251: Typo (missing spacebar) „body, a …“
Line 357: Typo (missing spacebar) „plate, mx…“

After correcting minor ambiguities, I am very pleased to suggest the manuscript to be accepted for the journal.

---

## Round 0.2 · Minor Revisions

Thank you for adressing our comments and suggestions. particularly adding information on apomorphic features linking previous finds to parasitic beetles. Your manuscript is as good as accepted. I just found some minor issues which i would you to address for publication:

Line 49: it would help to clarify or mention an example of what you consider a relatively small insect - what is the smallest insect that has been documented as a host

Link 65: thank you for integrating this new reference (Nagler and Haug 2015), but please also add it to the references

Line 262: add space between "body," and a

Line 273: please also take this opportunity to thank the reviewers.

---

## Author Rebuttal · Round 0.2

**FRIEDRICH-SCHILLER-UNIVERSITÄT JENA**

**Institut für Zoologie und Evolutionsforschung**
mit Phyletischem Museum,
Ernst-Haeckel-Haus und Biologiedidaktik

PD Dr. Hans Pohl

Erbertstrasse 1
07743 Jena

Telefon:     0 36 41 9-49156
Telefax:     0 36 41 9-49142
E-Mail:      hans.pohl@uni-jena.de
             www.speziellezoologie.uni-jena.de

Universität Jena · Institut für Zoologie und Evolutionsforschung

To

Dr. Kenneth De Baets

September 13th, 2018

Dear Kenneth,

Thank you very much for your efforts handling our manuscript and for your detailed comments and suggestions to improve our manuscript.

We addressed the comments by you and the reviewers individually below. In some cases we did not agree to your points or the points of the reviewers but provided a reasoning for our position.

**Detailed responses to your comments and the comments of the reviewers** (Comments in red, our response in black):

**Your comments:**

Lines 22–24: This is quite speculative. You present a single find demonstrating that this strategy of extant (=crown-group) strepsipteran is already present in at least on taxon in the Cretaceous, which is cool.

Due to limited fossil evidence or without ancestral state reconstruction, it is unclear what the larvae of early representatives of strepsiptera look like. However, it is not a stretch to consider that stem-group larvae could have a mixture of characters of both sister-groups and potentially even characters not seen in either lineage today.

I think rephrasing this as "Previous reports of Cretaceous crown-group strepsipteran immatures are more ambigious. They might  represent larvae of stem-group representatives or more likely parasitic beetle larvae."

We do not agree with this comment. Our primary larva is very likely a stem-group larva of Strepsiptera. No representatives of the Strepsiptera crown-group are known from the Cretaceous and therefore it would be extremely unlikely that a primary larva of a representative of the crown-group would be found.

The representatives of Strepsiptera from the Cretaceous found so far are without exception males. They are typical strepsipterans and differ only in a few details from the most "ancestral" extant representatives. Then why should Cretaceous primary larvae differ conspicuously from those of the "ancestral" extant Strepsiptera primary larvae?

You are right, of course, that we don't know how early representatives of Strepsiptera primary larvae look like. But we have explained that it is extremely unlikely that extremely aberrant larvae will occur

at the same time without fossil evidence for more aberrant adult male strepsipterans. We therefore prefer to maintain this sentence as it is.

Line 36: This is also reviewed/discussed in
Nagler, C., and Haug, J. T., 2015, From Fossil Parasitoids to Vectors: Insects as Parasites and Hosts: Advances in Parasitology, v. 90, p. 137-200.
This reference also discussed different kind of fossil evidence for parasitism/parasitoids you are discussing and highlight the importance of direct evidence for parasitic nature of particular ontogenetic stages.
I feel it would to reader to highlight that parasitic larval stages are often inferred based on presence of adults and only rarely based on direct evidence.
Many thanks for the hint to this reference. Since this sentence only refers to the age of the Strepsiptera, this quotation would make no sense for us at this passage. But we added it to the text further below.

Lines 47–48: Please provide a reference for this statement; Otherwise, i suggest to rewrite this as: "This size reduction suggest that females produce a huge number of ... "
We agree with this comment and added the reference (Pohl & Beutel 2008).

Line 57: Replace "reveals" with "suggest"
We agree with this comment and have replaced it.

Lines 64–65: I suggest to add a sentence that the presence of parasitic larvae are often inferred based on presence of adults attributable to particular lineages, but that direct evidence is better (e.g., Nagler and Haug, 2015).
We agree with this comment and rephrased the following sentences:
As with most cases of parasitism documented in the fossil record (e.g. Nagler & Haug, 2015), Mesozoic Strepsiptera provide only indirect evidence of this specialization. Based on the small size of adult males (ca. 1.5–3 mm), it was assumed, that the species were already endoparasites of other insects (Pohl & Beutel, 2008). However, conclusive evidence by direct detection of definite strepsipteran larvae was lacking, and therefore direct evidence for an endoparasitic life style of the immature stages of Mesozoic strepsipterans. Previously, endoparasitism was only documented by fossil representatives of Stylopidia from Eocene deposits, the strepsipteran subgroup with permanently endoparasitic females (Kinzelbach & Pohl, 1994; Henderickx et al., 2013).

Lines 76–77: This needs to be more clear: i suggest to write: are reported from Dominican amber (Poinar 2004), which is currently attributed to the Miocene: 15-20 mya (Itturalde-Vinent & MacPhee 1996; Itturalde-Vinent 2001).
Iturralde-Vinent, M. A., 2001, Geology of the amber-bearing deposits of the greater Antilles: Carribean Journal of Science, v. 37, no. 3-4, p. 141-166.

We agree with this comment and rewrote the sentence according to your suggestion.

Line 82: i guess you mean extant "crown-group" Strepsiptera" in this context. Can you rule out that it is a stem-group representative of Strepsiptera showing a mixture of characters of its sister-lineage?
Yes, we can rule out that these larvae are not stem-group representatives of Strepsiptera. This is discussed in detail in Beutel et al. (2016). Very likely these larvae are belonging to a cucujiform group of Coleoptera with parasitic habits, based on the following features: e.g. the lobe-like terminal elements of the legs and the pattern of setae on the abdomen.

Lines 82–84: I would be helpful to state which characters speak for this assignment.
We modified this sentence: The "planidium" from the Upper Cretaceous amber (Kathirithamby et al., 2017) is most likely a parasitic beetle larva based on the following features: small size (ca. 0.53 mm), conical head, absence of distinctly developed stemmata, coarse ctenidia, large pretarsal adhesive pad, absence of abdominal segment XI and terminal bristles (Batelka et al., accepted) (see below).

Lines 98–99: Yes, but ideally their placement should be accompagnied by phylogenetic/cladistic analysis and ancestral state reconstructions.
We agree with this comment on principle. However, a cladistics analysis with many ambiguous and/or missing characters would not be reliable in this case.

Line 105: Replace "Burmite" with "Burmese amber"
We agree with this comment and replaced "Burmite" with "Burmese amber".

Lines 121–122: I am aware about the technical difficulties, but is there no way to use computed tomography or other non-destructive techniques to reveal more about these specimens.
The extremely miniaturized primary larva is embedded in a relatively large piece of amber (26 x 22 x 10 mm) and therefore the resolution of micro-CT is too low, as in micro-CT the maximum resolution and the sample size are coupled (the smaller the sample the higher the resolution) (see e.g. Wipfler, B., Pohl, H., Yavorskaya, M.I., Beutel, R.G., 2016. A review of methods for analysing insect structures — the role of morphology in the age of phylogenomics. Current Opinion in Insect Science 18, 60–68.).

Line 219: maybe, but you only have evidence in this lineage; who says all strepsiptera larvae looked the same in the Cretaceous or even before that time. Maybe a substantial parts of lineages went extinct only after the Cretaceous. You would need more evidence to make such a claim. I would turn it around, you shows that this strategy of modern strepsiptera was already establishment in the Cretaceous. It helps little to interpret potential stem-group strepsipteran or coleoptera.
I would be more convinced if you would find evidence for such a strategy in the earlier strepsiptera. If they really originated in the earliest Permian 298.9, there is still a lot of time left to not show evolutioanry stability.

We agree with this comment and added: "A major point demonstrated her is that *this* Cretaceous larva of Strepsiptera differs only in minimal details from extant immatures …"

Lines 227–228: This is quite arm-waving and not necessary (if you have characters assigning those to parasitic beetles). Again, why do all strepsipteran primary larvae have to look the same; 300 Mya is a long time for various stem-group lineage to go extinct. Furthermore, this bears on the ability to discover and identify stem-group lineages of extant lineages which are highly morphologically derived. This can only be adequately done if you have a robust phylogeny of extant and fossil forms and morphologies of their larval stage to infer their ancestral state.

You're right, but as we explained above it is extremely unlikely that extremely aberrant strepsipteran larvae will occur at the same time without the occurrence of more basal adult male Strepsiptera as we they are known so far.

Lines 235–236: ok, so this means they could also represent a now extinct lineage of parasitic beetle larvae or could they even be a stem-group representative of strepsiptera? It would be good to be more clear which character can be used to unambigiously assign them to beetles.

We agree with this comment and added: All observed features are compatible with a placement in polyphagan beetles. The presence of coarse ctenidia and a pad-like pretarsal adhesive device are apomorphic features linking it clearly with beetle larvae very likely belonging to the cucujiform family Ripiphoridae (Beutel et al., 2016; Batelka et al., accepted).

**Reviewer 1**

Lines 155–156: In lines 155-156, there is no need to note that wings or wing buds are missing. No evidence of these would be expected in a strepsipteran larvae or any larvae that might be mistaken for this order, such as Meloidae, Rhipiphoridae, …

We agree with this comment and deleted this sentence.

Line 244: The term hatching in line 244 in not appropriate. The larvae emerge from the female after hatching internally from an egg. The term in this sentence should be changed to emerging.

We agree with this comment and changed the term to emerging.

Lines 243–246: In more than one place in the manuscript it is suggested that there must be a large number of primary larvae produced by each female but this appears to be speculation and if included should be identified as such. Based on knowledge of current strepsipterans, this could be true but the authors should not rely so heavily on this interpretation as fact. The reality is that we know nothing of the natural history of these larvae or their numbers from this geological age. Unless the unlikely event of the discovery of a gravid female of this age it may always be speculation as to the relative number of offspring. Likewise, lines 243-246 are stated as fact but is speculation for the species covered in this study.

We do not agree with this comment. We have phrased it carefully and not as a fact. However, in extant Strepsiptera the females are always clearly larger or at least the same size as the males. If one assumes a minimum size of 1.3 mm of the conspecific female, as the smallest described Mesozoic strepsipteran male, then a high number of extremely miniaturized primary larvae (approx. 200 μm) is very likely. It is therefore extremely unlikely that only a few primary larvae were produced.

**Reviewer 2**

Line 19: Mesozoic origin – it could be „at least" the Mesozoic origin
We agree with this comment and added 'at least'.

Line 21: „one of the most" or the most?
What we wrote is correct. The Bahiaxenidae are more basal than the Mengenilidae, but from this family only a single adult male is known.

Line 42: Bahiaxenidae probably also leave their host – is it supported by references?
The Bahiaxenidae are the sister group of all the remaining extant Strepsiptera (Bravo et al. 2009). Since the Mengenillidae are characterized by plesiomorphic features such as e.g. free-living females, it is very likely that the females of the Bahiaxenidae are also free-living.

Line 50: It is not clear, whether labrum is completly missing, or it is „largely reduced" (written in Lines 163-166) / or fused with other mouthparts – „lacking separate labrum" (Lines 204, 205), so I would prefer "not present as separate element".
We agree with this comment and have made this consistent as suggested by the reviewer ("not present as separate element").

Line 56: „enable the minute larvae to jump very efficiently" – not true for the family Stylopidae, so I would kindly suggest "for the most" of families"
We agree with this comment and changed the sentence to: "They enable the minute larvae to jump very efficiently, an ability secondarily lost in the primary larvae of the Stylopidae (Pohl & Beutel, 2005)."

Line 84: I am not familiar with an abbreviation „s.b."
We changed the abbreviation to "see below".

Line 85-87: Is the „planidium" always „legless"? Aren't also first instar larvae of Meloidae (with legs) reffered as „planidium"? This term is derived from the Greek word, which should mean „wanderer", so I assume the term is not such a problem in case of Strepsiptera.
According to Askew (1971) and Stehr (1991) the term "planidium" is only appropriate for legless larvae (parasitic Diptera or Hymenoptera).

Lines 87-94: Although I agree with neutral term „primary larva" for first instars of Strepsiptera, term „triungulinid" (= triungulin-like) also would not be such a problem for case of Strepsiptera, because it is not the same as „triungulin" (without -id).
As we have explained in this paragraph, we prefer the neutral term "primary larva".

Line 101: I would suggest to add the final years again – "endoparasitism is extended back by ca. 50 milion years to minimum age of ca. 100 mya"
We agree with this comment and changed the sentence accordingly.

Line 106: Better to use mya consistantly than „Ma"
We prefer "Ma"

Line 163: (st in Figs. 1A, B, 3A)
We agree with this comment and changed the sentence accordingly.

Line 187: Sternites IV-VI with two setae, not Sternites III-IV
We agree with this comment and changed it accordingly.

Line 197: Mengenilla chobauti
We agree with this comment and changed it accordingly.

Line 251: Typo (missing spacebar) „body, a …"
We agree with this comment and changed it accordingly.

Line 357: Typo (missing spacebar) „plate, mx…"
We agree with this comment (Line 395) and changed it accordingly.

We hope that we addressed all comments in a comprehensible way. If there are any questions or comments, please do not hesitate to contact me.

Sincerely yours,

(Hans Pohl)

---

## Round 0.3 · accepted · Accept

Thank you addressing my final comments and adding the missing reference. Looking forward to seeing you manuscript published.

# There was one final requirement raised by the Section Editors - specifically, they feel that the reference to the future deposition of the specimen should be removed (to avoid confusion, and in case the deposition doesn't happen as planned). Specifically, they request that the following text be removed from the manuscript: "and will be deposited in the Three Gorges Entomological Museum, Chongqing, China after 2027" (present at lines 13-14, pg 5, Materials and Methods). You can work with our Production group to remove this text.

---

## Author Rebuttal · Round 0.3

**FRIEDRICH-SCHILLER-UNIVERSITÄT JENA**

**Institut für Zoologie und Evolutionsforschung**
mit Phyletischem Museum,
Ernst-Haeckel-Haus und Biologiedidaktik

PD Dr. Hans Pohl

Erbertstrasse 1
07743 Jena

| Telefon: | 0 36 41 9-49156 |
| Telefax: | 0 36 41 9-49142 |
| E-Mail: | hans.pohl@uni-jena.de |
| | www.speziellezoologie.uni-jena.de |

Universität Jena · Institut für Zoologie und Evolutionsforschung

To

Dr. Kenneth De Baets

October 6th, 2018

Dear Kenneth,

Thank you so much for your positive message. That makes us happy that only very few things are left to be done.

**Line 49: it would help to clarify or mention an example of what you consider a relatively small insect - what is the smallest insect that has been documented as a host**

We agree with this comment and rephrased the sentence: "This size reduction enables the female to produce a huge number of offspring and the minute primary larvae are able to penetrate relatively small insect hosts such as Delphacidae (Auchenorrhyncha) with an adult size of 1.5–6 mm (Pohl & Beutel, 2008)."

**Link 65: thank you for integrating this new reference (Nagler and Haug 2015), but please also add it to the references**

Thank you very much for the hint that we forgot to add this reference to the references. I have added it now.

**Line 262: add space between "body," and a**

We agree with this comment and changed it accordingly.

**Line 273: please also take this opportunity to thank the reviewers.**

We agree with the comment and have added a sentence in which we thank you and the two reviewers.

I would like to thank you again for your very valuable comments and remarks.

Sincerely yours,

Hans Pohl